# Brain MRI super-resolution using 3D generative adversarial networks

Irina Sánchez , Verónica Vilaplana

Universitat Politècnica de Catalunya - BarcelonaTech
Department of Signal Theory and Communications
Barcelona, Spain
`veronica.vilaplana@upc.edu`

## Abstract

In this work we propose an adversarial learning approach to generate high resolution MRI scans from low resolution images. The architecture, based on the SRGAN model, adopts 3D convolutions to exploit volumetric information. For the discriminator, the adversarial loss uses least squares in order to stabilize the training. For the generator, the loss function is a combination of a least squares adversarial loss and a content term based on mean square error and image gradients in order to improve the quality of the generated images. We explore different solutions for the upsampling phase. We present promising results that improve classical interpolation, showing the potential of the approach for 3D medical imaging super-resolution.

## 1   Introduction

In many medical applications, high resolution images are required to facilitate early and accurate diagnosis. However, due to economical, technological or physical limitations, it may not be easy to obtain images at the desired resolution. Super-resolution techniques solve this problem by creating a High Resolution (HR) image from a Low Resolution one (LR). In the past decade a variety of super-resolution methods have been successfully applied to magnetic resonance imaging (MRI) data to increase the spatial resolution of scans after acquisition has been performed. Approaches can be broadly categorized into reconstruction-based and learning-based methods [1]. Within the second group, solutions based on deep learning are currently being investigated motivated by the success of deep learning models in many computer vision tasks [2, 3]. In particular, Generative Adversarial Networks (GANs) [4] are a very promising approach for image generation, and have been also used for super-resolution [5]. Recently, different architectures and loss functions that try to improve the quality of the images generated using GANs have been presented [6, 7, 8]. However, these approaches have been proposed for 2D data.

In this work we propose an architecture for MRI super-resolution that completely exploits the available volumetric information contained in MRI scans, using 3D convolutions to process the volumes and taking advantage of an adversarial framework, improving the realism of the generated volumes. The model is based on the SRGAN network [5]. The adversarial loss uses least squares to stabilize the training and the generator loss, in addition to the adversarial term contains a content term based on mean square error and image gradients in order to improve the quality of the generated images. We explore three different methods for the upsampling phase: an upsampling layer which uses nearest neighbors to replicate consecutive pixels followed by a convolutional layer to improve the approximation, sub-pixel convolution layers as proposed in [9] and a modification of this method [10] that alleviates checkbock artifacts produced by sub-pixel convolution layers [11].

1st Conference on Medical Imaging with Deep Learning (MIDL 2018), Amsterdam, The Netherlands.

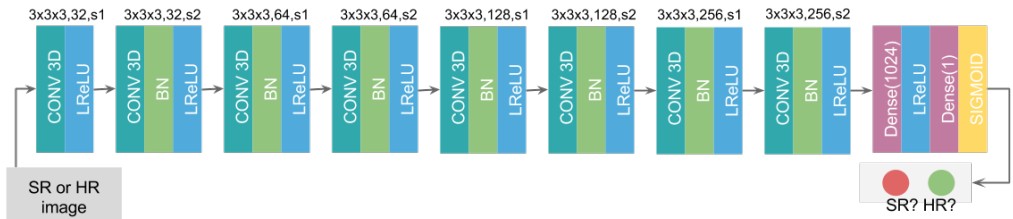

Figure 1: Architecture of the Discriminator network. For each convolutional layer: kernel size (3x3x3), number of filters, stride (s).

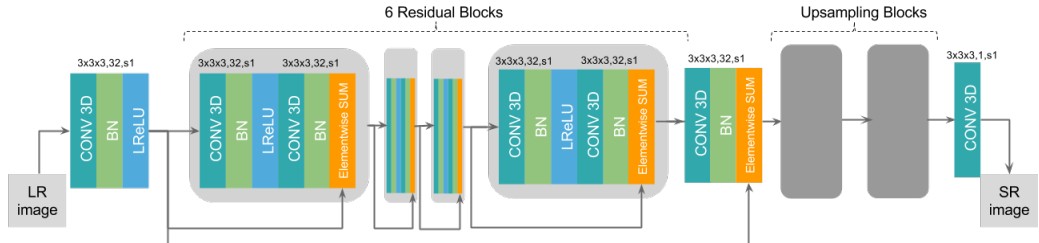

Figure 2: Architecture of the Generator network. For each convolutional layer: kernel size (3x3x3), number of filters, stride (s).

## 2 Method

### 2.1 Generative Adversarial Networks

GANs[4] are generative models that consist of a generator $G$ and a discriminator $D$ that compete in a two-player minimax game. The target of $G$ is to learn the distribution over data $x$ starting from sampling input variables from a uniform or Gaussian distribution $p_z(z)$, while the discriminator $D$ is typically a binary classifier that tries to decide whether a sample is from the training data or has been generated by $G$.

The two players learn by means of an adversarial training, where G has to learn how to cheat D, making the images perceptually closer to the input data, while D has to recognize efficiently the real samples from the fake ones. The process is formulated with the following minimax objective:

$$\min_G \max_D V(D, G) = \mathbb{E}_{x \sim p_{data}(x)}[\log D(x)] + \mathbb{E}_{z \sim p_z(z)}[\log(1 - D(G(z)))] \tag{1}$$

In the case of image super-resolution, the goal is to generate a high resolution image $I_{SR}$ from a low resolution input image $I_{LR}$. The image $I_{LR}$ is a low-resolution version of a high-resolution image $I_{HR}$, obtained by applying a Gaussian filter and a downsampling operator with downsampling factor $r$. The high-resolution images $I_{HR}$ are only available during training. The generator $G$ is a convolutional neural network that is trained to generate a high resolution counterpart from a low resolution input image. The discriminator $D$ is another neural network that tries to differentiate the generated $I_{SR}$ from the real $I_{HR}$.

Due to the adversarial formulation GANs may be difficult to train; it is necessary to provide a balance between both players, so neither of them can outperform the other. For that reason, different methods and architectures have been proposed recently to make GANs more stable, and also to increase the quality of the images. Examples of these methods, that we incorporate in our model, are the following: the use of batch normalization in all layers (except in the G output and D input), one-side label smoothing to prevent extreme extrapolation behavior in the discriminator and reduce its confidence and the use of loss functions that avoid the vanishing gradient problem of the classical approach and helps to stabilize the training.

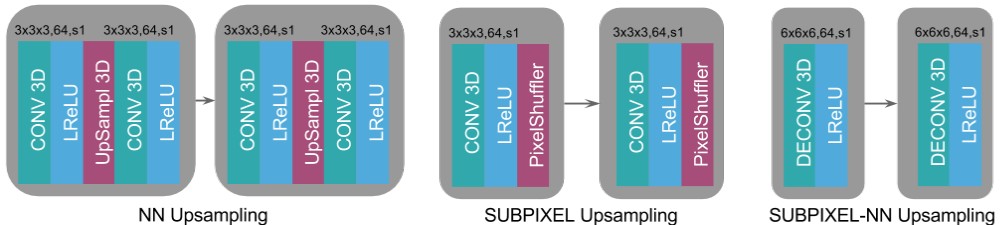

Figure 3: Architecture of the different upsampling methods.

## 2.2 Architecture

Our network architecture (Fig. 1 and Fig. 2) is based on the SRGAN model [5]. The main difference is found in the convolutional filters; in our architecture, to be able to deal with volumetric information we use 3D convolutional layers. Furthermore, we introduce changes in the loss function, both to make the network more stable and to improve the quality of the generated images. These changes will be explained in section 2.4.

The discriminator network is composed of eight convolutional layers with 3x3x3 kernels. The number of filters increases by a factor of 2 from 32 to 256. Strided convolutions are used to reduce the image resolution each time the number of features is doubled. After the convolutional layers, there are two dense layers and a final sigmoid activation that outputs a probability indicating whether the input image is real or fake.

For the generator, we use six residual blocks composed of a convolution with 32 filters of size 3x3x3, batch normalization, LeakyReLU activation and another convolution with the same parameters and batch normalization. As in the SRGAN network, there are connections between input and output in the residual blocks, and between the input of the first residual block and the output of the last residual block, allowing the network to access low level features and improve the quality of the generated images. After the residual blocks, there are upsampling blocks that increase the resolution of the input image. Each block up-samples the image by a factor of 2, so blocks are replicated in order to obtain higher upsampling factors. We explore different configurations for the upsampling blocks that are explained in the following section.

## 2.3 Upsampling methods

Figure 3 shows the configuration of the different methods used for the upsampling blocks in the generator network. In the following, we describe each low-resolution volume by a real valued tensor of size $H \times W \times D \times C$, and the high-resolution volume by $rH \times rW \times rD \times C$, where $r$ is the downsampling factor, $H$, $W$, $D$ and $C$ are the height, width, depth and number of channels, respectively.

### 2.3.1 Resize convolution

The first method upscales the input feature maps using nearest neighbor interpolation, and then employs a standard convolutional layer with both input and output in the higher resolution space.

### 2.3.2 Sub-pixel convolution

An efficient way to increase the resolution of a 2D image was proposed in [12]. In their work, the method to perform the upsampling operation consists of a convolutional layer followed by a pixel-shuffler that arranges the result of the convolution with shape $H \times W \times Cr^2$ into a tensor of size $rH \times rW \times C$. In our case, since we are dealing with 3D volumes, the size of the initial tensor is $H \times W \times D \times Cr^3$ and is finally arranged into a tensor of size $rH \times rW \times rD \times C$. This method is equivalent to a deconvolution operation with stride 2 as observed in [13]. Moreover, sub-pixel convolutions can be interpreted as a standard convolution in low-resolution space followed by a periodic shuffling operation.

Following this interpretation, we create a set of kernels that, rearranged, will build the HR image. Given a volume of size $H \times W \times D \times C$, convolved by a set of kernels $(nf, C, k, k, k)$ (where $nf$

is the number of filters of size $k \times k \times k$) after computing the pixel-shuffling operation we have a volume of size $rH \times rW \times rD \times \frac{nf}{r^3}$. Here, note that by rearranging pixels the feature maps (output channels after the convolution) are mixed, creating consecutive pixels.

The advantage of this method over resize convolution is that with the same computational complexity, it has more parameters, improving the modeling power of the network. However, it suffers from checkerboard artifacts since consecutive pixels depend on different feature maps that are independently randomly initialized.

### 2.3.3 Convolution nearest neighbor resize

A modification of the sub-pixel convolution was proposed in [10] in order to remove the checkerboard artifacts produced by random initialization while maintaining the number of trainable parameters. Resize convolution produces upsampled images free of checkerboard artifacts. For that reason, if the sub-pixel operation is made identical to a nearest neighbor resize after the initialization step, the advantages of the sub-pixel method will be preserved, while eliminating the undesired artifacts.

The method consists on performing first a deconvolution operation with stride 2 (equivalent to pixel-shuffling after convolution) with kernels initialized with size $(nf, \frac{nf}{r^3}, k, k, k)$. Then, an upsampling of this initialization is performed to finally compute the deconvolution layer with filters of size $(nf, \frac{nf}{r^3}, kr, kr, kr)$.

### 2.4 Loss function

A very critical point when designing a GAN is the definition of the loss function. Many methods have been proposed recently to stabilize the training and improve the quality of the synthesized images.

One of these works is [8], where the behavior of the sigmoid cross entropy loss function, commonly used in the classic GANs framework is studied; this loss function causes the problem of vanishing gradients for the samples that lie far away from the correct decision boundary. To overcome this problem a loss function based on least-squares is proposed (LSGAN), substituting the cross-entropy loss by a least square function with binary coding (1 for real, 0 for fake).

Using this formulation, equation 1 changes to:

$$\min_D V_{LSGAN}(D) = \frac{1}{2}\mathbb{E}_{x \sim p_{data}(x)}[D(x) - 1^2] + \frac{1}{2}\mathbb{E}_{z \sim p_z(z)}[D(G(z)^2] \tag{2}$$

$$\min_G V_{LSGAN}(G) = \frac{1}{2}\mathbb{E}_{z \sim p_z(z)}[(D(G(z) - 1)^2] \tag{3}$$

**Discriminator loss:** In our super-resolution model, the adversarial loss used for the discriminator is

$$l_D^{adv} = \frac{1}{2}[D(I_{\text{HR}}) - 1]^2 + \frac{1}{2}[D(G(I_{\text{SR}})]^2 \tag{4}$$

**Generator loss:** The loss function used for the generator is a combination of an adversarial term and a content term, as proposed in [5] (in our experiments we use $\alpha = 10^{-3}$):

$$l_G = \alpha * l_G^{adv} + l_G^{cont} \tag{5}$$

The adversarial loss is based on least squares:

$$l_G^{adv} = \frac{1}{2}[D(G(I_{\text{SR}}) - 1]^2 \tag{6}$$

while the content loss is a combination of two terms $l_G^{cont} = l_G^{cont/MSE} + l_G^{cont/gdl}$

The first term is the mean squared error (MSE) between the original high resolution image $I_{HR}$ and the super resolved image $I_{SR}$, calculated as

$$l_G^{cont/MSE} = \frac{1}{WHD}\sum_{x=1}^{rW}\sum_{y=1}^{rH}\sum_{z=1}^{rD}(I_{x,y,z}^{HR} - I_{x,y,z}^{SR})^2 \tag{7}$$

The pixel-wise MSE loss help to achieve high PSNR values, but tends to create blurry images. In order to improve the quality of the samples, an image gradient-based loss term is used as proposed in [14].

$$l_G^{cont/gdl} = ||\nabla I_{\text{HR}x}| - |\nabla I_{\text{SR}x}||^2 + ||\nabla I_{\text{HR}y}| - |\nabla I_{\text{SR}y}||^2 + ||\nabla I_{\text{HR}z}| - |\nabla I_{\text{SR}z}||^2 \quad (8)$$

This Gradient Difference Loss (GDL) sharpens the image prediction by penalizing the differences of image gradient predictions.

# 3  Experiments and Results

We perform our experiments using a set of normal control T1-weighted images from the Alzheimer's Disease Neuroimaging Initiative (ADNI) database (see www.adni-info.org for details). Skull stripping is performed in all volumes and part of the background is removed. Final volumes have dimensions 224x224x152.

Due to memory constraints the training is patch-based; for each volume we extract patches of size 128x128x92, with a step of 112x112x76, so there are 8 patches per volume, with an overlap of 16x16x16. We have a total number of 589 volumes, 470 are used for training while 119 are used for testing. We use batches of two patches, thus for each volume we perform 4 iterations. We use Adam optimization for both the generator and the discriminator, while the learning rate for the generator is set to 1e-5 and for the discriminator 1e-4.

To evaluate the quality of the images synthesized by our model we make two sets of experiments, using the original MRI volumes as ground truth and training the network with downsampled versions of the images by factors 2 and 4, using the three upsampling strategies described in Section 2.3. In the following, Subpixel stands for sub-pixel convolution and Subpixel-NN stands for convolution nearest neighbor resize. We have also compared the performance of our model with a classical cubic spline interpolation.

| | Upsample x2 | | | | | | | |
| | Cubic Int. | | Resize Conv. | | Subpixel | | Subpixel-NN | |
| | PSNR | SSIM | PSNR | SSIM | PSNR | SSIM | PSNR | SSIM |
|---|---|---|---|---|---|---|---|---|
| **Mean** | 38.06 | 0.9848 | 39.11 | **0.9913** | 39.09 | 0.9898 | **39.28** | 0.9849 |
| **Std** | 1.2085 | 0.0020 | 1.0608 | 0.0014 | 1.0203 | 0.0016 | 1.0724 | 0.0028 |
| **Min** | 34.65 | 0.9792 | 35.93 | 0.9868 | 36.61 | 0.9855 | 36.65 | 0.9781 |
| **Max** | 41.45 | 0.9897 | 41.88 | 0.9940 | 42.39 | 0.9933 | 42.54 | 0.9907 |
| | Upsample x4 | | | | | | | |
| | Cubic Int. | | Resize Conv. | | Subpixel | | Subpixel-NN | |
| | PSNR | SSIM | PSNR | SSIM | PSNR | SSIM | PSNR | SSIM |
| **Mean** | 31.76 | 0.9412 | 33.33 | **0.9688** | 32.86 | 0.9638 | **33.58** | 0.9582 |
| **Std** | 0.9948 | 0.0078 | 1.1813 | 0.0070 | 1.2241 | 0.0085 | 1.1456 | 0.0097 |
| **Min** | 29.78 | 0.9312 | 30.54 | 0.9531 | 29.96 | 0.9462 | 31.01 | 0.9388 |
| **Max** | 33.74 | 0.9534 | 36.86 | 0.9816 | 36.51 | 0.9787 | 37.23 | 0.9770 |

Table 1: Numerical Results

The generated volumes are compared to the ground-truth volumes in terms of peak signal-to-noise-ratio (PSNR) and structural similarity index measure (SSIM). Numerical results can be found in Table 1. We observe that the resize convolution upsampling method outperforms the others in terms of SSIM a metric that is closer to human visual perception than the PSNR, while sub-pixel nearest neighbor convolution is the best in terms of PSNR.

In Figures 4 and 5 we present some results of the different methods, for downsampling factors 2 and 4. If we take a closer look at the resulting images, we can see that the resize convolution method produces perceptually better results. While the changes proposed by the sub-pixel nearest neighbor convolution method improve the behavior of the original sub-pixel convolution, this upsampling method still produce some checkboard artifacts in our images.

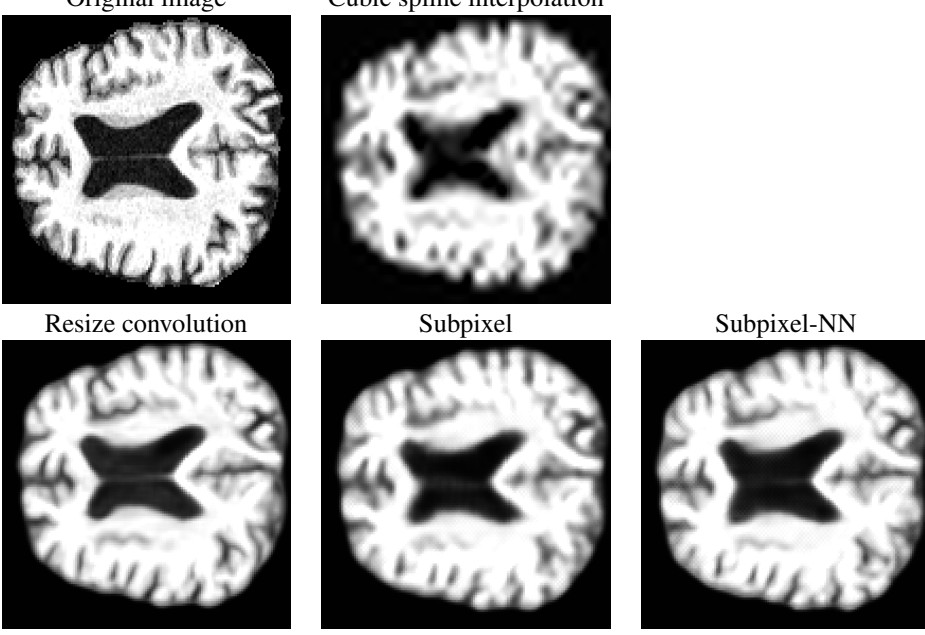

Figure 4: Illustration of SR results using a downsampling factor of 4. The first row shows the original high resolution image and the result of cubic spline interpolation. The next row presents the results of applying our GAN with the three proposed upsampling methods: resize convolution, sub-pixel and subpixel-NN.

## 4   Conclusions

In this work a method for MRI super-resolution has been implemented within the generative adversarial framework. We use an adversarial term loss in the generator loss to create more realistic samples, and a content loss to reduce the differences between real and generated images. We present promising results that improve classical interpolation when the downsampling factor is high, showing the potential of the approach for 3D medical imaging super-resolution. Possible future work involves the exploration of better architectures and the inclusion of other perceptual terms in the loss function in order to increase the quality of the generated volumes. Further experiments need to be done using other datasets and comparing the performance of this approach to other methods. Also a mean opinion score (MOS) test should be performed to evaluate the ability of the method to generate perceptually convincing images. This is an important issue since minor errors in the reconstruction might lead to big differences in clinical interpretation.

**Acknowledgments**

This work has been partially supported by the project MALEGRA TEC2016-75976-R financed by the Spanish Ministerio de Economía y Competitividad and the European Regional Development Fund (ERDF).

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
