# OpenReview forum: "Brain MRI super-resolution using 3D generative adversarial networks"
_MIDL.amsterdam/2018/Conference — MIDL 2018 Poster_

### Review · AnonReviewer3 · 2018-04-27
**Review on Brain MRI super-resolution using 3D GANs**

**Rating:** 3
**Confidence:** 2

**Review:**

This paper extends a recently proposed an adversarial learning approach to generate high resolution MRI scans from low resolution volumes.  The approach explored is the SRGAN [5], which combines a GAN loss with a reconstruction loss and can estimate images for 4x upscaling factors.  Similarly, this proposed paper explores the same idea to estimate Brain MRIs for 4x upscaling factors.  The main contribution of this paper is the extension of SRGAN for 3D, which introduces some difficulties in the learning and inference processes.  Another contribution of the paper is the testing of different upsampling methods: resize convolution,  sub-pixel convolution,  convolution nearest neighbor resize.  Results rely on a dataset containing 589 volumes, where 470 are used for training and 119 are used for testing.  The quantitative measures (SSIM, PSNR) show quite competitive values, particularly in comparison with bicubic interpolation (this is the only baseline method).  However, the qualitative results show that the estimated volumes do not present visually sharp results.

In general, I think the paper explores a methodology that is useful and seems to be on the right track.  It still needs some more work to become a more solid paper, such as: more baseline results, mean opinion score (MOS) evaluation, and visually sharper reconstructions.  In addition, the reproducibility of the work can be improved with a detailed explanation of how the HR reconstructed patches are combined in the overlapping regions of 16^3.

Minor issues:
- Are the training and testing volumes from different patients? Please clarify.
- In Figure 4, please show the LR input images.



**Special Issue:**

No

---

### Review · AnonReviewer2 · 2018-05-09
**A very good paper**

**Rating:** 4
**Confidence:** 3

**Review:**

Overall:
The goal of the paper is to use adversarial training for brain MRI super-resolution. The authors use a specific architecture of a neural network for the super-resolution task. Further, they consider different upsampling schemas. Eventually, the authors compare the results on the brain MRI scans. I like the paper and I believe that it provides an interesting insight into the topic.

Strengths:
+ The paper is easy to follow and all concepts are explained in a lucid manner.
+ The methods are properly chosen and motivated.
+ The experiments are performed properly.
+ The presented results are interesting and show that the resize convolutions seem to be the most promising.

Remarks:
* Minor
- According to the LaTex style, the authors' names should be in bold.
- In Eq. (5), I presume the star corresponds to standard multiplication. If so, I suggest to remove it because a star is often used to denote the convolution.
- Did the experiment was repeated couple of times? If so, it would be beneficial to provide standard errors to further see whether the differences among considered upscaling schemas are significant.

**Special Issue:**

Yes

---

### Review · AnonReviewer1 · 2018-05-09
**This paper presents the generation method of high resolution images from low resolution MRI images using generative adversarial networks. Although the various upsampling methods have attempted and solved the problem, the experimental results are somewhat marginal.**

**Rating:** 3
**Confidence:** 2

**Review:**

Quality & Clarity

#1. This paper is well organized, and the upsampling ideas were clearly explained.
#2. The description of experimental results is well written.

Originality & Significance

(+) Authors have proposed various upsampling methods to integrate generative adversarial networks.
(-) Why did they use generative adversarial networks? If both low and high resolution images are given, another approaches such as super-resolution networks would be helpful.

**Special Issue:**

No

---

### Decision · Program_Chairs · 2018-05-15
**Paper114 Acceptance Decision**

Poster